# Influence of Phenotypes on the Metabolic Syndrome of Women with Polycystic Ovary Syndrome over a Six-Year Follow-Up in Brazil

**DOI:** 10.3390/biomedicines11123262

**Published:** 2023-12-09

**Authors:** Jose Maria Soares-Jr., Sylvia Asaka Yamashita Hayashida, Jose Antonio Miguel Marcondes, Gustavo Arantes Rosa Maciel, Cristiano Roberto Grimaldi Barcellos, Giovana De Nardo Maffazioli, Karla Krislaine Alves Costa Monteiro, Jose Antonio Orellana Turri, Ricardo Azziz, Edmund Chada Baracat

**Affiliations:** 1Laboratório de Ginecologia Estrutural e Molecular (LIM-58), Disciplina de Ginecologia, Departamento de Obstetrícia e Ginecologia, Hospital das Clínicas HC-FMUSP, Faculdade de Medicina, Universidade de São Paulo, São Paulo 05403-010, Brazil; sylvia.hayashida@gmail.com (S.A.Y.H.); garmaciel@gmail.com (G.A.R.M.); giovanamaffazioli@hotmail.com (G.D.N.M.); karlakrislane_@hotmail.com (K.K.A.C.M.); antonioturri@usp.br (J.A.O.T.); ecbaracat@usp.br (E.C.B.); 2Divisão de Endocrinologia, Hospital das Clínicas HC-FMUSP, Faculdade de Medicina, Universidade de São Paulo, São Paulo 05403-010, Brazil; marcondesmd@uol.com.br (J.A.M.M.); endocrinologia.barcellos@gmail.com (C.R.G.B.); 3Academic Health and Hospital Affairs, The State University of New York (SUNY) System Adminstration, Buffalo, NY 14261, USA; razziz67@gmail.com

**Keywords:** polycystic ovary syndrome, metabolic syndrome follow-up, phenotypes

## Abstract

Background: We followed polycystic ovary syndrome (PCOS) women with metabolic syndrome (MS) over a six-year treatment period and evaluated the influence of PCOS phenotypes on MS and on the risk for type 2 diabetes mellitus (T2DM). Methods: This was an observational study of 457 PCOS women, whose demographic, clinical, hormonal, and metabolic data underwent analysis. The PCOS women were divided into four groups per NIH recommendations. Results: After a follow-up of a mean of six years (1–20 years), 310 patients were selected to assess the development of T2DM and MS. The clinical and biochemical parameters, along with the Rotterdam phenotypes, were evaluated. Data were analyzed using Student’s *t*- and the Pearson chi-square tests for data variation and group proportions, respectively. Additionally, multivariate analysis was applied to evaluate the effect of PCOS phenotypes on the risk for MS and T2DM. Patients of the four PCOS phenotypes did not differ in age, body mass index, total testosterone, insulin resistance, and dyslipidemia, but phenotype A patients showed the highest risk for T2DM. A decrease in androgen levels was not followed by an improved metabolic profile; instead, there was a significant increase in the number of T2DM cases. Conclusion: Phenotype A women are at the highest risk for type 2 diabetes mellitus.

## 1. Introduction

Polycystic ovary syndrome (PCOS) is a heterogeneous disorder characterized by chronic anovulation and hyperandrogenism, and it affects approximately 5% to 15% of unselected women of reproductive age [1,2,3,4]. PCOS is associated with metabolic dysfunction and is a risk factor for the development of type 2 diabetes mellitus (DM), cardiovascular diseases (CVDs), and endometrial cancer [3,4,5]. Insulin resistance (IR) has been regarded as the link between dysfunctional carbohydrate metabolism and increased risk of CVD events, as well as other metabolic disorders in PCOS [6].

The NIH consensus panel in 2012 [1] advocated for the sole use of the broader 2003 Rotterdam criteria for diagnosis [2]. It also recommended the identification of the four PCOS phenotypes according to clinical or biochemical hyperandrogenism, ovulatory dysfunction, and polycystic ovary morphology [1,3,4]. 

Some PCOS studies with long-term follow-ups [5,6,7,8,9,10,11,12] depict worsening glucose intolerance (GI) leading to heightened rates of conversion from normal glucose levels to GI or T2DM, mainly in obese women [5,6,13]. Thus, periodic T2DM screening for PCOS women is recommended [14,15,16]. PCOS is considered an independent risk factor for T2DM [14,15,16], but the inherent risk of each of its phenotypes is not totally known.

Metabolic syndrome in PCOS has been hypothesized as a progressive disorder; however, confirmatory follow-up data for it are scarce. A study by Carmina et al. [9] involved follow-up visits of 193 patients every 5 years over a 20-year span. The patients showed improvement in hyperandrogenism and oligo/anovulation features; nonetheless, metabolic disorders persisted, especially associated with an increase in abdominal circumference. Hence, fat tissue may have a role in the metabolic disorders of PCOS. Recently, another study confirmed a reduction in PCOS features in the fourth decade of life [12]. However, the researchers did not assess the influence of PCOS phenotypes on the parameters. Moreover, the impact of PCOS phenotypes on the treatment and the effect of the treatment on the long-term risk of MS remains unclear [9,10,11,12]. Therefore, data on those parameters are necessary to understand the metabolic disorder and its development in PCOS patients. Also, a follow-up of the treatment may help us to understand the real risk for PCOS women. The aim of the present study was to evaluate the clinical, hormonal, and metabolic features of women undergoing long-term treatment and follow-up for PCOS to better understand the progression of MS, as well as the influence of different phenotypes of PCOS on the risk of MS and T2DM. 

## 2. Materials and Methods

### 2.1. Subjects and Design

This is a longitudinal study of the retrospective cohort type. The sample consisted of 608 consecutive hyperandrogenic women from the Disciplina de Ginecologia, Departamento de Obstetrícia e Ginecologia and the Divisão de Endocrinologia, Departamento de Clínica Médica, Hospital das Clínicas, Faculdade de Medicina, Universidade de São Paulo. The patients were selected from 1994 to 2018 medical files. The recruitment was conducted by referring patients who came to primary health care clinics in the São Paulo public system.

Inclusion criteria: diagnosis of PCOS according to the Rotterdam criteria [2] and age between 18 and 40 years. Exclusion criteria: idiopathic hirsutism, hyperprolactinemia, hypothyroidism, late-onset congenital adrenal hyperplasia, androgen-producing tumors, Cushing disease; use of hormonal contraceptives in the previous three months and of anti-androgens in the previous six months; and pregnancy and puerperium. Disorders causing hyperandrogenism were ruled out with specific tests. 

The control group consisted of healthy women aged between 18 and 40 years. They all had regular menstrual cycles (between 24 and 38 days) and hormonal profiles within the normal range for the age group. Participants had taken no hormonal or metabolic medications for at least 3 months before inclusion in the study. 

### 2.2. Clinical Features Database and Complementary Exams

The database with the clinical electronic medical records stored information from the first and the follow-up visits. The hyperandrogenic women referred to our service were assessed according to a care protocol and the data were registered in the medical records. The clinical history included the following: menstrual features, acne, hirsutism, alopecia, and clitoromegaly. The clinical evaluation consisted of anthropometric measures, with an investigation of clinical signs of hyperandrogenism using the Ferriman–Gallwey score and the features of virilism [17]. 

Measurements of the following hormones were taken at all visits: follicle-stimulating hormone (FSH), luteinizing hormone (LH), estradiol (E2), prolactin (PRL), total and free testosterone, 17-hydroxyprogesterone (17OHP), dehydroepiandrosterone sulfate (DHEAS), androstenedione, thyroid-stimulating hormone (TSH), and free thyroxine (T4). The metabolic evaluation included lipid and glucose profiles. The collection was always carried out in the morning. We collected from the fifth to the eighth day of the menstrual cycle, in the follicular phase; in amenorrhea, this was at any time, but we always took care to measure concomitant progesterone to exclude collection in the ovulatory phase. Ultrasound evaluation of the ovaries was performed, and they were deemed polycystic if they had 12 or more follicles with diameters of 2 to 9 mm [18]. In addition, the used ultrasound machine was a Power Vision 7000 (Toshiba Medical Systems Corporation, Tochigi, Japan) equipped with 3.5 MHz and 7.0 MHz wide-band transducers for the abdominal and transvaginal routes, respectively. The same gynecologist read each ultrasound scan.

After the baseline exams, the women underwent the adrenal-stimulation test with synthetic ACTH (cortrosyn^TM^) to exclude late-onset congenital adrenal hyperplasia [19], as well as the glucose tolerance test [20]. The 250 mcg of cortrosyn was administered intravenously in the follicular phase of the menstrual cycle. After 60 min, values above 15 ng/mL were a confirmed diagnosis of congenital adrenal hyperplasia (CAH). Between 10 and 15 ng/mL, a genetic test was performed to investigate CYP21A2 gene mutation.

In women with very high testosterone levels (2.5× above the normal standards of the method), the evaluation of the tumor was performed. If the cut-off of testosterone levels was greater than 200 ng/dL, a CT scan of the upper abdomen or MRI of the pelvis was applied to exclude adrenal or ovarian tumors, respectively. If the results were negative, the ovarian suppression test was carried out with the use of a gonadotropin-releasing hormone (GnRH) analog. When suppression was insufficient (no drop of more than 50% of baseline levels in serum total testosterone levels after the test), the adrenal depression test was conducted with 1 mg of dexamethasone. In this way, women with probable concomitant adrenal production were detected [21]. 

The oral glucose tolerance test (OGTT) was performed according to ADA recommendations [20]. Impaired glucose tolerance (IGT) was defined as two-hour glucose levels of 140 to 199 mg/dL (7.8 to 11.0 mmol) on the 75 g OGTT. Plasma glucose concentration was determined by the glucose oxidase method, and glycated hemoglobin (HbA1C) by hemoglobin glycosylation per a method certified by the NGSP-US (National Glycohemoglobin Standardization Program) [22]. Total cholesterol, HDL-c and triglycerides were assessed using enzyme methods (Roche Laboratories), and LDL-C was estimated with the Friedwald formula.

For hormone analyses, progesterone was measured by immunofluorometric assay (Wallac, Turku, Finland) using Auto DELFIA kits; androstenedione, PRL, LH, and FSH were measured by immunofluorometric assay; and insulin, 17OHP, DHEAS were measured by radioimmunoassay (Cisbio International, Saclay, France and DSL, Austin, TX, USA). The testosterone and SHBG levels were measured using immunofluorometric assay (Wallac, Finland) up to October 2012. After this date, the assay was replaced with the electrochemiluminescent immunoassay (Modular, Roche, Basel, Switzerland). Consequently, the normal ranges changed as follows: a) total testosterone: 14 to 98 ng/dL and 14 to 48 ng/dL for the former and the latter kit assays, respectively; and b) free testosterone: 2 to 45 pmol/L and 2.4 to 37 pmol/L. Free testosterone was calculated by Vermeulen’s formula. Testosterone and free testosterone values were normalized for data comparison. All analyses were performed twice, and the intra-assay and inter-assay coefficients of variation did not exceed 10% and 15%, respectively. 

### 2.3. Visits

In the first year, PCOS women were seen every 4 months for follow-up treatment until their condition stabilized; then, they were seen every 6 months to every year, depending on their needs. Laboratory tests were repeated every 6 months. Blood was collected for insulin determination with OGTT every year. The choice of treatment was made according to the women’s objective: reproductive desire, treatment of hirsutism, or menstrual control. For PCOS women with MS, adequate medication (statin, metformin, or antihypertensive drugs) was prescribed. For this study, we used the initial data and the data from the last query. Drugs were not suspended for the last evaluation. In addition, anthropometric parameters such as waist circumference (WC) and body mass index (BMI) were evaluated in each visit. The WC measurements were taken halfway between the lowest rib and the top of the hipbone, and the BMI was calculated using the following formula: BMI = kg/m^2^, where kg is a person’s weight in kilograms and m^2^ is their height in meters squared. The systemic arterial blood pressure (BP) was measured in each visit.

An evaluation was carried out for the clinical and laboratory parameters of the following: (a) phenotypes of women with PCOS; (b) clinical parameters; (c) hormonal parameters; (d) metabolic parameters; (e) insulin sensitivity parameters; (f) prevalence of MS; (g) prevalence of DM diagnosis; and (h) cardiovascular risk parameter [VAI].

As evaluated by Guastela et al. [3] and reinforced by the NIH recommendation [1], there are four PCOS phenotypes: type A (hyperandrogenism, chronic anovulation, and polycystic ovaries); B (hyperandrogenism and chronic anovulation without polycystic ovaries); C (hyperandrogenism and polycystic ovaries); and D (chronic anovulation and polycystic ovaries without hyperandrogenism).

The Matsuda index (ISI Matsuda) was calculated using the formula described by Matsuda and Di Fronzo [23]: 10,000/(G0 × I0 × Gmean × Imean)1/2. Also, we applied the Homeostasis Model Assessment-Insulin Resistance (HOMA-IR) model, based on fasting glucose (mmol/L) × fasting insulin (mUI/mL)/22.5 [24], and the quantitative insulin sensitivity check index (QUIKCKI)—1/[logI0(uU/mL) + log G0(ng/dL)]—for evaluating insulin resistance [25].

The visceral adiposity index (VAI) was defined by the following formula: VAI = WC/36.58 + (1.89 × BMI) × (TG/0.81) × 1.52/HDL. It was described by Amato et al. [26], where TG (triglycerides) and HDL levels are expressed in mmol/L. It is used as a marker for cardiovascular risk related to adipose tissue distribution and function. For a diagnosis of metabolic syndrome (MS), three of the five following NCEP-ATP III criteria should be presented in the patient: (a) WC > 88 cm; (b) triglycerides ≥ 150 mg/dL; (c) HDL-C < 50 mg/dL; (d) blood pressure ≥ 130 mmHg × 85 mmHg; and (e) basal glucose ≥ 100 mg/dL [27]. Diabetes was diagnosed by the same parameters as previous studies [20,28]. 

Metformin was prescribed for patients with IR or glucose intolerance. Patients with neither MS nor reproductive desire used a hormonal contraceptive along with anti-androgen drugs for menstrual irregularity and hirsutism. 

The adverse events of treatment and metabolic, hormonal, and clinical aspects were evaluated during follow-up. Only women who did not miss any outpatient service visits for at least 12 months were included and women taking more than 80% of the prescribed drugs, which were quantified by the clinic notes in the medical records for checking the adherence of patients to treatment. All overweight and obese women were advised to start a healthy, hypocaloric diet (the goal daily calorie intake was 1400 kcal) and to engage in daily physical activities, such as walking and aerobic exercises. We controlled the lifestyle parameters in all visits.

### 2.4. Statistical Analysis

Clinical and demographic data were presented by mean, median, standard deviation, interquartile range, total frequency, and percentage. Previously to statistical tests, distribution analyses were performed using Shapiro–Wilk and Smirnov Kolmogorov tests. Continuous data were compared between groups using Student’s *t*-test, ANOVA, Mann–Whitney or Wilcoxon, and categorical data were compared with chi-square or Fisher tests, when appropriate. Additionally, Pearson or Spearman correlation and linear regression were calculated. Statistical analysis was performed using STATA 16 SE. A significance level of 5% was used in all tests.

## 3. Results

### 3.1. Baseline Characteristics

Of the 457 consecutive PCOS women initially included at baseline, 145 were excluded from follow-up for one or more of the following reasons: (a) poor compliance (n = 64); (b) inadequate follow-up (n = 54); (c) hormonal contraceptive use (n = 22); (d) uncontrolled DM type 2 at baseline (n = 2); and (e) familial hypertriglyceridemia (n = 1). Two additional patients dropped out due to thrombosis after oral contraceptive therapy. However, investigation showed that one had thrombophilia (protein C deficiency) and the other had phospholipid syndrome. The final number of women was 310 (Figure 1). 

The PCOS group was younger than the control group, had a significantly higher degree of hirsutism, and had significant values of WC (*p* < 0.05) (Table 1). The mean levels of LH and LH/FSH were higher than in the control group (*p* < 0.01). The levels of total and free testosterone, 17OHP, and androstenedione in the PCOS group were superior to those of the control group (*p* < 0.01), but the mean levels of FSH and SHBG were lower in the PCOS group (*p* < 0.01). Most of the metabolic parameters were worse in the PCOS group than in the control group. The high BP, TG, and IGT or DM; low HDL-C; and wide WC and MS were more frequent in the PCOS group than the control (*p* < 0.05). The total number of participants with IGT and DM as well as the markers of glucose metabolism were higher in the PCOS group (*p* < 0.01); the VAI score presented high values compared to the control group (*p* < 0.001). The number of women with a BMI in the normal range was larger in the control group than in the PCOS group (*p* = 0.015) (Table 1). 

Spearman’s correlation of the baseline variables in the PCOS group revealed a strong and positive correlation of WC and WC/H (H = height) with BMI, and the ISI Matsuda presented a positive and significant correlation with QUICKI (*p* < 0.001). The BMI had a moderate and positive correlation with fasting insulin and HOMA-IR (*p* < 0.001), and such was the correlation of WC and WC/H with fasting. The correlation of BMI values was moderate and negative with QUICKI (*p* < 0.001). The WC, WC/H, and G120 showed a moderate and negative correlation with QUICKI and Matsuda (*p* < 0.001). Total and free testosterone did not significantly correlate with clinical and metabolic disorders of PCOS (Figure 2 and Figure 3).

The results of the linear regression models between the WC and the clinical and biochemical characteristics of both the PCOS and the control groups are in Table 2. The PCOS women’s WC was positively associated with age, MS, DM, and total cholesterol (*p* < 0.05). The comparison between the PCOS group and the controls revealed that, in both groups, WC was significantly and positively associated with BMI, GI, serum levels of LDL-C, TG, fasting glucose, 2 h post-load glucose, fasting insulin, HOMA-IR, and VAI and negatively associated with HDL-C, SHBG levels, and QUICKI.

### 3.2. Follow-Up Characteristics

One PCOS woman with MS died owing to complications from bariatric surgery (pulmonary embolism). Three overweight women and one from the obese group lost weight and improved their PCOS clinical features.

Clinical and biochemical evaluations were carried out during the last follow-up visit. The percentage of PCOS patients using medications was the following: (a) 45.2% used oral hormonal combined contraceptives (OHCC); (b) 19% used OHCC associated with anti-androgens, metformin, or drugs for metabolic correction, such as hypotensors, hypoglycemics, statins, and anorectics; (c) 26.1% used metformin alone or in association with statins for metabolic correction; (d) 20.6% used no drugs throughout the study due to a reproductive desire or intolerance to metformin or contraceptives; and (e) 8.1% used statins, ciprofibrate, levothyroxine, anti-androgens, or psychotropics. The linear regression did not find a significant influence on the metabolic profile.

Table 3 describes the data on the hormonal and metabolic features of PCOS women and compares the baseline and the follow-up data after a mean of six years. A significant reduction took place in the mFG score and the androgenic hormone profile (testosterone, free testosterone, DHEAS, and androstenedione), whereas there was an increase in the follow-up values of SHBG levels (*p* = 0.001) in participants who used hormonal contraceptives combined with anti-androgen drugs. In the PCOS group, a decrease in the number of normal-weight women and an increase in the number of obese women were found after follow-up (*p* < 0.05). Hyperandrogenemia decreased during the follow-up compared to baseline. However, the reduction in the prevalence of MS was not significant. The frequency of T2DM and the VAI score worsened. The percentage of T2DM at baseline and at the last follow-up was 3.9% and 11.6%, respectively. The percentage of metabolic syndrome at baseline and at the last follow-up was 29.1% and 25.1%, respectively. 

Table 4 shows the data of women with PCOS according to phenotype during follow-up. At baseline, the BMI of phenotype D was significantly lower than that of phenotype B (*p* < 0.05). The mFG score of phenotype C was significantly higher than that of phenotypes B and D (*p* < 0.01). Phenotype A had the highest total and free testosterone levels (*p* < 0.05). Phenotype C had an increased incidence of adrenal hyperandrogenemia (increased DHEAS values). Among the obese, the percentage of women with phenotype D was the lowest (*p* < 0.01). Fasting plasma glucose and triglycerides had lower values in phenotype D women than in those with phenotypes A and B (*p* < 0.01). There were no differences in WC, SHBG, insulin sensibility, and dyslipidemia (except TG, which was significantly higher in phenotype A than D among the four phenotypes). During follow-up, the degree of hirsutism reduced significantly in phenotypes A, B, and C compared to their baseline values and phenotype D. In phenotypes A and B, the follow-up androgen profile showed a significant decrease in 17OHP, testosterone, free testosterone, DHEAS, and androstenedione (*p* < 0.001). The SHBG values in the follow-up data in each group were higher than those at baseline. The number of MS cases diminished in phenotype A during follow-up compared to baseline (*p* = 0.015). The number of phenotype A women with fasting GI also dropped during follow-up compared to baseline (*p* = 0.001), but the number of women with T2DM increased significantly in phenotype A (*p* < 0.001). Also, only in phenotype A was there a significant increase in VAI values (*p* < 0.01). 

Figure 4 depicts the baseline demographics and clinical results in the PCOS phenotype and control groups. The BMI of phenotype D was significantly lower. The WC was significantly wider in phenotype B than in the control group; the values of T were significantly higher in all phenotypes. The SHBG values of phenotype B were significantly lower than those of the control group. Phenotype C had significantly higher values of 17OHP. The DHEAS values were higher in phenotype C, but the difference had no statistical significance. Glucose was significantly higher in phenotype A. The 2 h post-load glucose levels were higher in phenotypes A and B. The Matsuda test result was significantly lower in phenotype A compared to other phenotypes. The VAI was significantly higher in phenotype A than in the control group.

Table 5 displays the linear regression involving VAI and the clinical and biochemical features of the women with PCOS according to phenotype after follow-up. There was a positive correlation of VAI with fasting glucose intolerance and with T2DM in phenotypes A (*p* < 0.001) and C (*p* = 0.025). The VAI correlated positively with fasting glucose intolerance, two-hour post-load glucose, fasting insulin, insulin overload (120′), HbA1c, and HOMA-IR. 

## 4. Discussion

PCOS is known as a hormonal and metabolic disorder [29,30] as well as an independent risk factor for T2DM [14,15,16]. Our data confirm this fact. Despite the treatment, there was no reduction in the incidence of MS and T2DM in phenotype A of PCOS. In addition, MS frequency did not differ among the phenotypes, except for phenotype D, which had the lowest risk. Similar results were found by Jamil et al., 2016 [30]. The weight profiles likely influenced the metabolic risk in the phenotypes. Obesity was difficult to reduce in our sample and may justify the reason for treatment failure. 

Phenotypes A and B were also at a higher risk of cardiovascular diseases during follow-up, including metabolic disorders and weight gain. Cardiovascular disease is silent, and endothelial dysfunction may take years [31]. Such a long time span hinders follow-up and acts as a limitation on our study, which was conducted to follow up women with PCOS—the group that causes the most concern in clinical practice. 

Another factor that contributes to cardiovascular disease is T2DM. Recent long-term follow-up studies have shown that the risk for developing hypertension, MS, prediabetes, and T2DM is higher in PCOS women under 40 years of age [32,33]. The evidence provided by our study is in accord with such studies, as our data were derived from a population whose mean age was 32.5 years and whose progression to T2DM was significant. 

Metabolic syndrome is a major health concern due to the attendant CV risks and their sequelae (acute myocardial infarction and stroke) [34]. Patients with PCOS are affected by metabolic disorders and are at a high risk of CV diseases. During follow-up, the VAI values worsened in phenotype A patients, the very women who run a higher risk of CV diseases even when receiving guidance and medication. Therefore, independently of therapeutic measures, the aggravation of metabolic conditions tends to persist. Unfortunately, the consolidated literature on follow-up data is lacking [9,10,11,12]. Hence, our data on this issue can be crucial for drawing up the necessary strategies for each phenotype, and most of all, for phenotype A. In addition, other authors found a high risk of diabetes and increased cardiovascular disease risk factors among 199 Chinese women with PCOS in the long-term follow-up study (mean 10 years), but they failed to identify the specific PCOS phenotype. Also, the patients in those studies were older than ours [35]. Some changes in the metabolic therapy of PCOS patients may be required, particularly in terms of weight loss. 

The VAI is a novel sex-specific index based on WC, BMI, triglycerides, and HDL-C. It is used as a marker of CV risk related to adipose tissue distribution and function [36,37,38]. A study [38] showed that VAI may have a significant inverse correlation with insulin sensitivity during the euglycemic hyperinsulinemic clamp. Although not a perfect index [37,38], it is useful in assessing the magnitude of the risk of CV disease. In our data, VAI, together with the other parameters, showed that overweight patients were at an intermediate risk, and two of them developed diabetes. 

During follow-up, the VAI values worsened in phenotype A PCOS patients—the very women who run a higher risk of CV diseases even when receiving guidance and medication. Therefore, independently of therapeutic measures, the aggravation of metabolic conditions tends to persist. Unfortunately, the consolidated literature on follow-up data is lacking [9,10,11,12]. 

Besides metabolic disorders, another matter of concern is hirsutism, which is more prevalent in obese women with insulin resistance, as verified in the study conducted by Korhonen et al. [39] with the Finnish population. In our data, hirsutism did not accompany metabolic syndrome or obesity. The study possibly lacked the necessary power to detect the differences among phenotypes. Carmina et al. [9] found that phenotypic changes are age-related. In fact, the clinical features of phenotype A decrease with aging. One likely reason is that our patients were younger than those in the last study [9].

The association between serum testosterone and insulin resistance or metabolic syndrome discussed in some studies [40,41] was questioned by Corbould [42] due to the lack of data showing cause–effect associations. An increase in serum levels of DHEAS and 17OHP was detected in phenotype C, leading to the suspicion of an adrenal component in this phenotype. 

The PCOS women undergoing the conventional treatment (hormonal contraceptives and anti-androgens) in general showed improvement in hirsutism and in menstrual pattern but not in the metabolic component during follow-up. Metabolism remains a challenge, as shown by our cases of phenotypes A, B, and C during follow-up. The androgenic profile, including DHEAS, is possibly involved in heightening the metabolic risk [21]. Nevertheless, even the reduction in androgen levels with the treatment did not lead to a great improvement in carbohydrate metabolism disorders [39,43,44]. 

Metformin’s long-term benefits in women with PCOS are still a controversial point in the literature [45,46]. Despite the advice given to every patient about the importance of a healthy diet and physical activity, no significant weight loss was observed, especially among the diabetic patients. Perhaps this fact may shed some light on the cases of MS and DM after follow-up. Another possibility is the low adherence to the metformin treatment due to side effects; another is the outcome of the Glintborg et al. [47] study that reported considerable weight gain after 12 months of follow-up as a result of oral contraceptive pills. According to these authors, the effect of the oral contraceptive pill might interfere with the weight reduction of patients undergoing nutritional and physical therapy or receiving orientation.

Our study has a few limitations: (a) the number of women lost to follow-up (21.8%); (b) it was not possible to evaluate the real impact of treatment on phenotype changing due to the low number of participants in phenotype groups B, C, and D; and (c) two types of standard kits for testosterone analyses were used during follow-up. Among the strong points of this study, the T2DM diagnosis relied on the OGTT test (120′ stands out), corroborating the studies by Legro et al. (2005) [6] and Andersen and Glintborg [15]; but, (d) we were unable to manage a control group camp (in conditions) to use for a long time, due to lack of patient compliance; and (e) the age of PCOS patients was lower than the control patients. 

## 5. Conclusions

Our follow-up data suggest that PCOS patients has high risk for DM, MS and cardiovascular (VAI). The phenotype A has the highest risk for diabetes mellitus and tends to increase the VAI. Further studies are necessary to confirm our data. 

## Figures and Tables

**Figure 1 biomedicines-11-03262-f001:**
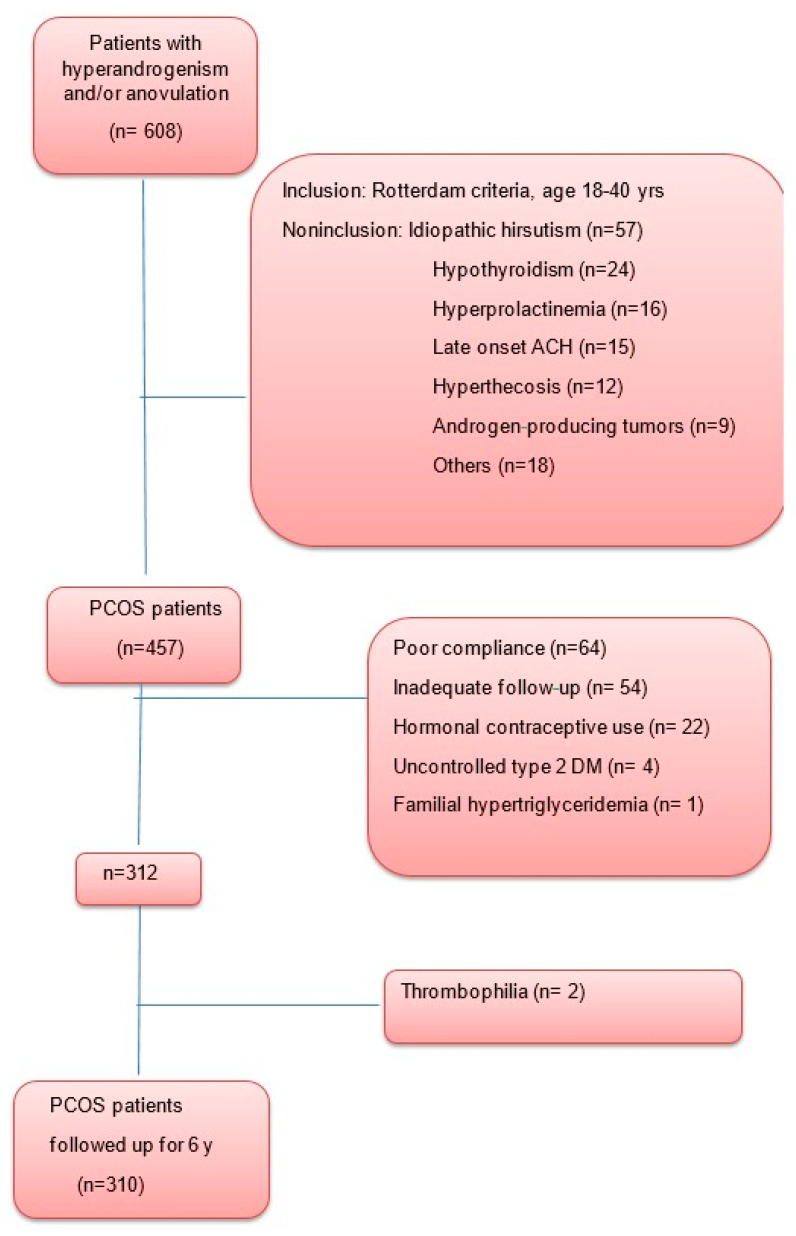
Study flowchart.

**Figure 2 biomedicines-11-03262-f002:**
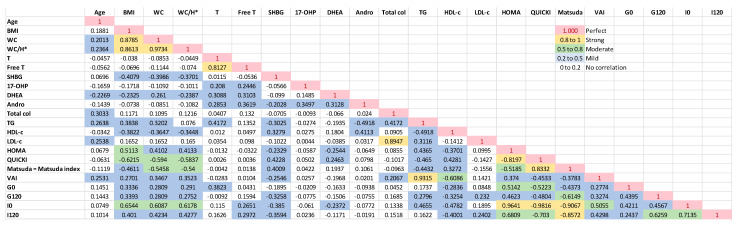
Spearman’s Correlation—Baseline PCOS. Free T (Free Testosterone) X T (Total testosterone) were not considered; HOMA and QUICKI X G0 (fasting plasma glucose) and I0 (fasting plasma insulin); VAI (visceral adipose index) X BMI (body mass index), WC (waist circunference), TG (triglycerides) and HDL–c (HDL cholesterol), as they are part of the calculation formula itself. SHBG = sex hormone binding globulin; 17–OHP = 17 OH progesterone; DHEAS = dehydroepiandrosterone sulfate; Andro = androstenedione; Matsuda = ISI Matsuda; G120 = 2 h OGTT glucose; I120 = 2 h OGTT insulin. * H = height.

**Figure 3 biomedicines-11-03262-f003:**
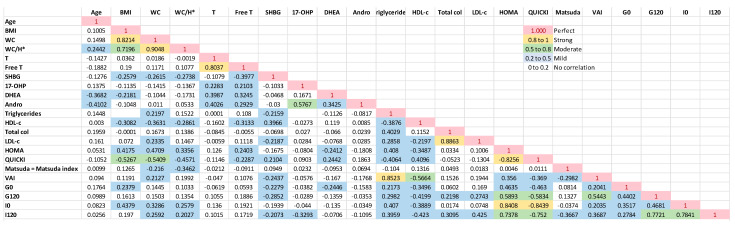
Spearman’s Correlation—Follow PCOS. Free T (Free Testosterone) X T (Total testosterone) were not considered; HOMA and QUICKI X G0 (fasting plasma glucose) and I0 (fasting plasma insulin); VAI (visceral adipose index) X BMI (body mass index), WC (waist circunference), TG (triglycerides) and HDL–c (HDL cholesterol), as they are part of the calculation formula itself. SHBG = sex hormone binding globulin; 17–OHP = 17 OH progesterone; DHEAS = dehydroepiandrosterone sulfate; Andro = androstenedione; Matsuda = ISI Matsuda; G120 = 2 h OGTT glucose; I120 = 2 h OGTT insulin. * H = height.

**Figure 4 biomedicines-11-03262-f004:**
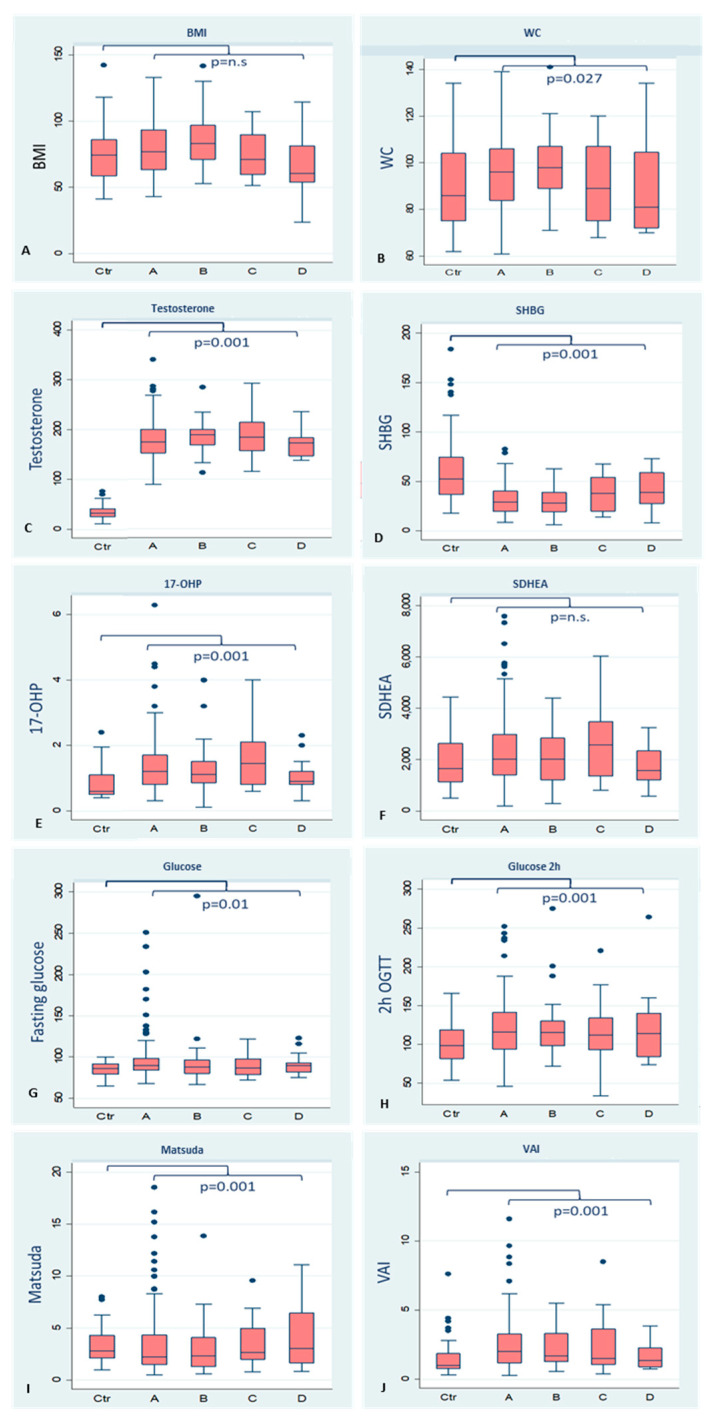
Baseline demographics and clinical exams in PCOS patients, control, and phenotypes groups. (**A**) (BMI): *p*-value = 0.023 between control and phenotype D; (**B**) (WC): *p*-value = 0.027 between control and phenotype B; (**C**) (Testosterone): *p*-value = 0.001 between control and phenotypes A, B, C and D; (**D**) (SHBG): *p*-value = 0.01 between control and phenotype B; (**E**) (17-OHP): *p*-value = 0.04 between control and phenotype C; (**F**) (SDHEA): *p*-value = n.s. (**G**) (Glucose): *p*-value = 0.03 between control and phenotype A; (**H**) (Glucose 2 h): *p*-value = 0.01 between control and phenotype A and *p*-value = 0.02 between control and phenotype B; (**I**) (Matsuda): *p*-value = 0.01 between control and phenotype A; (**J**) (VAI): *p*-value = 0.001 between control and phenotype A. Ctr = control; A = phenotype A; B = phenotype B; C = phenotype C; D = phenotype D; BMI = body mass index; WC = waist circumference; SHBG = sex hormone binding globulin; SDHEA= Dehydroepiandrosterone sulfate; 17-OHP = 17-OHP progesterone; Glucose = Fasting glucose; Glucose 2 h = overload after 75 g glucose. n.s. = non significative.

**Table 1 biomedicines-11-03262-t001:** Baseline clinical and biochemical characteristics of PCOS women and control group.

		PCOS (n = 310)	Control (n = 71)	*p*-Value
Clinical Characteristics	Age (years, mean ± SD)	25.5 ± 5.5	28.4 ± 5.4	0.001
	Menarche (years, mean ± SD)	12.6 ± 1.9	12.1 ± 2.2	ns
	BMI (kg/m^2^, mean ± SD)	30.6 ± 7.2	28.8 ± 6.5	ns
	WC (cm, mean ± SD)	94.6 ± 16.2	90.8 ± 7.0	0.027
	F-G Score [(median (Q1–Q3)]	12 (7–16)	2 (0–5)	0.001
Obesity Classification	Normal (%)	78/310 (25.2)	28/71 (39.4)	0.015
	Overweight (%)	86/310 (27.7)	14/71 (19.7)	0.053
	Obesity (%)	146/310 (47.1)	29/71 (40.9)	ns
Hormonal Profile	FSH (IU/L, mean ± SD)	5.13 ± 1.61	5.72 ± 1.88	0.002
	LH (IU/L, mean ± SD)	10.4 ± 6.2	5.75 ± 3.87	0.001
	LH/FSH (mean ± SD)	2.04 ± 1.05	1.07 ± 0.70	0.001
	E2 (pg/mL, mean ± SD)	64.4 ± 42.6	67.2 ± 44.6	ns
	17OHP (ng/mL, mean ± SD) *	1.33 ± 0.79	0.83 ± 0.40	0.001
	Testosterone (ng/dL, mean ± SD)	47.1 ± 24.1	34.8 ± 13.2	0.001
	SHBG (nmol/L, mean ± SD)	32.7 ± 15.5	60.8 ± 34.7	0.001
	Calculated free testosterone (pmol/L, mean ± SD)	32.3 ± 19.8	16.5 ± 8.9	0.001
	DHEAS (ng/mL, mean ± SD)	2268.6 ± 1289.7	1950.3 ± 1035.8	ns
	Androstenedione (ng/mL, mean ± SD)	3.45 ± 1.39	1.27 ± 0.45	0.001
Metabolic Syndrome Evaluation (ATP III)	Fasting glucose intolerance or T2DM (%)	45/303 (14.8)	1/71 (1.4)	0.01
	↑ BP (≥130 × 85 mmHg) (%)	68/305 (22.3)	6/71 (8.4)	0.001
	↑ TG (≥150 mg/dL) (%)	83/310 (26.8)	5/69 (7.2)	0.001
	↓ HDL (<50 mg/dL) (%)	182/310 (58.7)	23/69 (33.3)	0.001
	↑ WC (>88 cm) (%)	167/299 (55.8)	34/70 (48.6)	0.027
	Metabolic syndrome (%)	89/306 (29.1)	4/70 (5.7)	0.001
Evaluation of Metabolic Profile	Fasting glucose intolerance (G0′) (%)	45/301 (14.9)	1/71 (1.4)	0.01
	2 h OGTT intolerance (G120’) (%)	49/275 (17.8)	3/48 (6.2)	0.001
	HbA1C intolerance (%)	31/232 (13.4)	2/34 (8.8)	0.001
	Total of intolerance (%)	125/310 (40.3)	6/71 (8.5)	0.002
	DM (G0′) (%)	0 /301 (0.0)	0/71 (0.0)	ns
	DM (G120’) (%)	11/275 (4.0)	0/48 (0.0)	0.004
	DM (HBA1C) (%)	1/232 (0.43)	0/34 (0.0)	
	Total of T2DM (%)	12/310 (3.9)	0/71 (0.0)	0.004
Metabolic Profile	Total cholesterol (mg/dL, mean ± SD)	179.6 ± 36.4	177.1 ± 31.9	ns
	HDL-c (mg/dL, mean ± SD)	48.2 ± 14.0	56.5 ± 16.3	0.001
	LDL-c (mg/dL, mean ± SD)	108.1 ± 31.8	102.8 ± 28.0	ns
	TG (mg/dL, mean ± SD)	120.6 ± 71.2	94.7 ± 65.2	0.001
	Fasting glucose (G0′) (mg/dL, mean ± SD)	89.7 ± 9.7	85.7 ± 7.7	0.01
	2 h OGTT glucose (G120’) (mg/dL)	122.8 ± 39.6	101.1 ± 25.3	0.001
	Fasting insulin (I0′) (µU/mL, mean ± SD)	19.0 ± 14.9	12.7 ± 15.7	0.001
	2 h OGTT insulin (I120’) (µU/mL, mean ± SD)	162.1 ± 133.7	196.6 ± 48.9	ns
	HbA1C (%,mean ± SD)	5.51 ± 0.60	5.07 ± 0.38	0.001
	HOMA-IR (mean ± SD)	4.43 ± 3.79	2.39 ± 1.50	0.001
	QUICKI (mean ± SD)	0.324 ± 0.036	0.346 ± 0.04	0.001
	ISI Matsuda (mean ± SD)	3.380 ± 2.89	3.489 ± 2.09	ns
	VAI (mean ± SD)	2.378 ± 1.772	1.469 ± 1.225	0.001

Data presented as mean ± SD or Percentage (%); Mann–Whitney/Wilcoxon; chi-square/Fisher; * Adrenal enzymatic deficiency was excluded by cortrosyn test; 17OHP = 17OH progesterone; BP = blood pressure; TG = triglycerides; HbA1C = glycated hemoglobin. ns = non significative. ↑ = increased; ↓ = decreased.

**Table 2 biomedicines-11-03262-t002:** Linear regression between WC and clinical and biochemical characteristics of the PCOS and control groups.

		PCOS (n = 310)		Control (n = 71)	
		β (95% CI)	*p*-Value	β (95% CI)	*p*-Value
Clinical Characteristics	Age (years)	0.397 (0.186–0.608)	0.001	0.450 (−0349–1.249)	0.265
	Menarche (years)	–1.514 (–2.599–0.428)	0.006	−0.278 (−3.091–2.536)	0.843
	BMI (kg/m^2^)	2.009 (1.868–2.150)	0.001	2.112 (1.623–2.600)	0.001
Hormonal Profile	FSH IU/L)	0.086 (−1.253–1.426)	0.899	0.345 (−1.882–2.571)	0.758
	LH (IU/L)	–0.642(−0.965–0.320)	0.001	−0.431 (−1.531–0.669)	0.437
	E2 (pg/mL)	–0.015 (−0.069–0.040)	0.595	−0.033 (−0.132–0.066)	0.511
	17OHP (ng/mL) *	–1.874 (−4.535–0.787)	0.166	−5.746 (−15.272–3.780)	0.233
	Testosterone (ng/dL)	–0.005 (−0.028–0.018)	0.663	−0.163 (−0.486–0.161)	0.319
	SHBG (nmol/L)	–0.061 (−0.099–0.023)	0.002	−0.173 (−0.293–0.0523)	0.005
	Calculated free testosterone (pmol/L)	0.074 (−0.006–0.153)	0.069	0.436 (−0.035–0.907)	0.069
	DHEAS (ng/mL)	–0.003 (−0.004–0.001)	0.001	−0.002 (−0.006–0.002)	0.345
	Androstenedione (ng/mL)	–0.642 (−3.493–2.209)	0.653	4.956 (−5.347–15.260)	0.338
Metabolic syndrome evaluation (ATP III)	Metabolic syndrome (%)	7.365 (5.1076–9.622)	0.001	3.000 (−28.424–3.424)	0.781
Evaluation of Metabolic Profile	2h OGTT intolerance (G120’) (%)	20.632 (14.384–26.881)	0.001	22.030 (4.714–39.346)	0.013
	Total of DM (%)	8.969 (5.803–12.134)	0.001	7.359 (−7.557–22.276)	0.328
Metabolic Profile	Total cholesterol (mg/dL)	0.053 (0.007–0.100)	0.025	0.046 (−0.086–0.177)	0.490
	HDL-c (mg/dL)	–0.406 (−0.519–0.294)	0.001	−0.518 (−0.750-−0.287)	0.001
	LDL-c (mg/dL)	0.088 (0.036–0.139)	0.001	0.191 (0.047–0.334)	0.010
	TG (mg/dL)	0.054 (0.030–0.077)	0.001	0.119 (0.061–0.177)	0.001
	Fasting glucose (G0’) (mg/dL)	0.112 (0.028–0.196)	0.009	0.601 (0.076–1.125)	0.025
	2h OGTT glucose (G120’) (mg/dL)	0.100 (0.046–0.154)	0.001	0.261 (0.081–0.441)	0.005
	Fasting insulin (I0’) (µU/mL)	0.453 (0.356–0.551)	0.001	0.346 (0.087–0.606)	0.010
	HbA1C (%)	1.843 (−0.161–3.848)	0.071	25.250 (10.664–39.836)	0.001
	HOMA-IR	1.705 (1.307–2.104)	0.001	6.319 (3.909–8.728)	0.001
	QUICKI	–265.92 (−306.23–225.62)	0.001	−253.08 (−351.69–154.46)	0.001
	ISI Matsuda	−2.775 (−3.408–2.142)	0.001	0.491 (−3.158–4.140)	0.783
	VAI	2.414 (1.594–3.234)	0.001	7.157 (4.165–10.150)	0.001

Data presented as mean ± SD or Percentage (%); Mann–Whitney/Wilcoxon; chi-square/Fisher; * Adrenal enzymatic deficiency was excluded by cortrosyn test; 17OHP = 17 OH progesterone; BP = blood pressure; TG = triglycerides; HbA1C = glycated hemoglobin; CFT = calculated free testosterone.

**Table 3 biomedicines-11-03262-t003:** Clinical and biochemical characteristics of 310 PCOS women in a long-term follow-up study.

		Baseline (n = 310) Mean ± SD	Follow-Up (n = 310) Mean ± SD	*p*-Value
Clinical Characteristics	Age (years)	25.5 ± 5.5	32.5 ± 7.7	0.001
	BMI (kg/m^2^)	30.6 ± 7.2	31.3 ± 6.8	ns
	WC (cm)	94.6 ± 16.2	97.9 ± 12.6	ns
	F-G Score	11.9 ± 6.3	9.1 ± 5.4	0.001
Obesity Classification	Normal (%)	78/310 (25.2)	55/303 (18.2)	0.029
	Overweight (%)	86/310 (27.7)	84/303 (27.7)	ns
	Obesity (%)	146/310 (47.1)	164/303 (54.1)	0.025
Hormonal Profile	17OHP (ng/mL) *	1.33 ± 0.79	1.04 ± 0.78	0.001
	Testosterone (ng/dL)	47.1 ± 24.1	38.8 ± 26.9	0.001
	SHBG (nmol/L)	32.7 ± 15.5	76.7 ± 85.0	0.001
	Calculated free testosterone (pmol/L)	32.3 ± 19.8	21.0 ± 19.1	0.001
	DHEAS (ng/mL)	2268.6 ± 1289.7	1986.4 ± 1161.5	0.009
	Androstenedione (ng/mL)	3.45 ± 1.39	2.13 ± 1.28	0.001
Metabolic Syndrome Evaluation (ATP III)	Fasting glucose intolerance or DM (%)	45/303 (14.8)	65/305 (21.3)	ns
	↑ BP (≥130 × 85 mmHg) (%)	68/305 (22.3)	69/305 22.6)	ns
	↑ TG (≥150 mg/dL) (%)	83/310 (26.8)	88/302 (29.1)	ns
	↓ HDL (<50 mg/dL) (%)	182/310 (58.7)	154/302 (51.0)	0.018
	↑ WC (>88 cm) (%)	167/299 (55.8)	155/262 (59.2)	ns
	Metabolic syndrome (%)	89/306 (29.1)	77/307 (25.1)	ns
Metabolic Profile	Total cholesterol (mg/dL)	179.6 ± 36.4	181.2 ± 34.3	ns
	HDL-c (mg/dL)	48.2 ± 14.0	50.6 ± 14.4	ns
	LDL-c (mg/dL)	108.1 ± 31.8	107.3 ± 30.5	ns
	TG (mg/dL)	120.6 ± 71.2	125.6 ± 67.4	ns
	Fasting glucose (G0′) (mg/dL)	89.7 ± 9.7	93.8 ± 23.0	ns
	Fasting insulin (I0′) (µU/mL)	19.0 ± 14.9	19.4 ± 15.8	ns
	2 h OGTT insulin (I120’) (µU/mL)	162.1 ± 133.7	177.2 ± 172.6	ns
	HbA1C (%)	5.49 ± 0.56	5.62 ± 0.90	ns
	HOMA-IR	4.43 ± 3.79	4.31 ± 3.68	ns
	QUICKI	0.324 ± 0.036	0.323 ± 0.033	ns
	ISI Matsuda	3.380 ± 2.89	3.864 ± 3.327	ns
	VAI	2.378 ± 1.772	3.100 ± 2.214	0.003
		n/total (%)	n/total (%)	
Evaluation of Metabolic Profile	Fasting glucose ≥ 100 mg/dL (G0′) (%)	45/301 (14.9)	53/305 (17.4)	ns
	2 h OGTT intolerance (G120’) (%)	49/275 (17.8)	19/164 (11.0)	0.014
	HbA1C intolerance (%)	31/232 (13.4)	27/236 (11.4)	ns
	Total of intolerance (%)	125/310 (40.3)	99/310 (31.9)	ns
	T2DM (G0′) (%)	0 /301 (0.0)	12/305 (3.93)	ns
	T2DM (G120’) (%)	11/275 (4.0)	18/164 (11.0)	0.045
	T2DM (HBA1C) (%)	1/232 (0.43)	6/236 (2.54)	ns
	Total of T2DM (%)	12/310 (3.9)	36/310 (11.6)	0.031
Metabolic Syndrome Evaluation (ATP III)	Fasting glucose intolerance or DM (%)	45/303 (14.8)	65/305 (21.3)	ns
	↑ BP (≥130 × 85 mmHg) (%)	68/305 (22.3)	69/305 22.6)	ns
	↑ TG (≥150 mg/dL) (%)	83/310 (26.8)	88/302 (29.1)	ns
	↓ HDL (<50 mg/dL) (%)	182/310 (58.7)	154/302 (51.0)	0.018
	↑ WC (>88 cm) (%)	167/299 (55.8)	155/262 (59.2)	ns
	Metabolic syndrome (%)	89/306 (29.1)	77/307 (25.1)	ns

Data presented as mean ± SD or Percentage (%); Mann–Whitney/Wilcoxon; chi-square/Fisher; * Adrenal enzymatic deficiency was excluded by cortrosyn test; 17OHP = 17OH progesterone; BP = blood pressure; TG = triglycerides; HbA1C = glycated hemoglobin; WC = waist circumference. Ns = non-significant. ↑ = increased; ↓ = decreased.

**Table 4 biomedicines-11-03262-t004:** Clinical and biochemical characteristics of PCOS women according to the phenotypes in a long-term follow-up study.

		Phenotype A (n = 233)			Phenotype B (n = 36)			Phenotype C (n = 20)			Phenotype D (n = 21)		
		Basal Mean ± SD	Follow-Up Mean ± SD	*p*-Value	Basal Mean ± SD	Follow-Up Mean ± SD	*p*-Value	Basal Mean ± SD	Follow-Up Mean ± SD	*p*-Value	Basal Mean ± SD	Follow-Up Mean ± SD	*p*-Value
Clinical Characteristics	Follow-up (months)		86.8 ± 63.5			102.3 ± 63.6			67.6 ± 51.8			74.8 ± 55.3	-
	BMI (kg/m^2^)	30.7 ± 7.1	31.6 ± 6.8	ns	32.9 ± 7.8 *	32.4 ± 7.2	ns	28.9 ± 6.1	29.6 ± 6.4	ns	26.8 ± 6.4 *	28.0 ± 5.9	ns
	WC (cm)	94.7 ± 15.8	97.5 ± 11.8	ns	99.7 ± 16.0	100.5 ± 14.8	ns	90.4 ± 16.8	101.4 ± 20.0	ns	89.9 ± 22.9	96.0 ± 17.2	ns
	F-G score	12.4 ± 6.2 †	9.7 ± 5.3 †	0.001	11.2 ± 5.9 †^,^‡	7.7 ± 5.1†	0.002	15.1 ± 4.1 †^,^‡	11.8 ± 4.7 †	0.019	2.9 ± 2.0 †	2.8 ± 2.6 †	ns
Hormonal Profile	17OHP (ng/mL)	1.34 ± 0.80	1.01 ± 0.78	0.001	1.26 ± 0.78	1.26 ± 0.77	ns	1.55 ± 0.91	1.15 ± 1.05	ns	1.08 ± 0.48	0.85 ± 0.40	ns
	Testosterone (ng/dL)	51.6 ± 24.2 ^§^	41.2 ± 28.3 ^§^	0.001	38.4 ± 19.2 ^§^	32.8 ± 22.4 ^§^	0.013	35.5 ± 19.6 ^§^	32.8 ± 19.3 ^§^	ns	23.6 ± 9.9 ^§^	28.9 ± 19.2 ^§^	ns
	SHBG (nmol/L)	31.9 ± 14.7	76.8 ± 87.7	0.001	30.4 ± 15.5	74.4 ± 67.4	0.004	37.4 ± 18.1	78.5 ± 75.4	ns	40.8 ± 19.4	82.3 ± 98.6	ns
	Calculated free testosterone (pmol/L)	35.7 ± 19.6 ^§^	22.4 ± 20.0	0.001	28.7 ± 20.9 ^§^	15.8 ± 12.4	0.001	21.8 ± 15.2 ^§^	19.5 ± 19.2	ns	14.2 ± 7.4 ^§^	16.6 ± 18.6	ns
	DHEAS (ng/mL)	2301.4 ± 1311.6	1989.4 ± 1082.9	0.029	2107.1 ± 1173.2	1687.0 ± 1187.7	ns	2746.8 ± 1528.2	2607.0 ± 1780.3	ns	1768.2 ± 812.0	1744.5 ± 890.5	ns
	Androstenedione (ng/mL)	3.62 ± 1.38 ^||^	2.20 ± 1.34	0.001	2.78 ± 1.44 ^||^	1.82 ± 1.10	0.014	3.32 ± 1.22	1.92 ± 1.12	0.001	2.97 ± 1.07	2.33 ± 1.24	ns
Metabolic Profile	Total cholesterol (mg/dL)	179.0 ± 36.9	180.4 ± 35.0	ns	186.6 ± 33.3	181.0 ± 31.1	ns	184.8 ± 44.9	192.7 ± 39.4	ns	170.5 ± 27.7	175.9 ± 27.7	ns
	HDL-c (mg/dL)	47.9 ± 13.7	50.2 ± 14.5	ns	47.6 ± 13.5	51.0 ± 15.6	ns	49.8 ± 17.4	51.7 ± 12.4	ns	50.7 ± 14.2	51.8 ± 14.5	ns
	LDL-c (mg/dL)	106.9 ± 32.3	106.5 ± 31.1	ns	116.0 ± 27.8	109.8 ± 26.7	ns	112.3 ± 38.7	114.8 ± 34.9	ns	103.2 ± 25.2	102.6 ± 23.7	ns
	TG (mg/dL)	122.3 ± 66.3 ^#^	129.1 ± 69.4 ^#^	ns	115.4 ± 65.0	117.9 ± 69.7	ns	116.6 ± 70.7	122.4 ± 51.8	ns	82.7 ± 39.2 ^#^	103.4 ± 51.0 ^#^	ns
	G0′ (mg/dL)	89.6 ± 9.2	94.5 ± 21.7	ns	89.6 ± 11.6	93.7 ± 36.4	ns	90.0 ± 9.4	90.5 ± 14.7	ns	90.8 ± 12.2	89.9 ± 12.6	ns
	G120′ (mg/dL)	121.3 ± 37.7	141.8 ± 58.1	0.01	130.3 ± 47.9	130.6 ± 55.4	ns	120.0 ± 43.6	144.6 ± 60.8	ns	125.7 ± 47.3	115.8 ± 61.0	ns
	I0′ (µU/mL)	19.4 ± 15.3	20.5 ± 17.2	ns	21.5 ± 16.7	17.5 ± 11.4	ns	15.3 ± 9.8	122.4 ± 51.8	ns	14.1 ± 10.2	13.1 ± 8.2	ns
	I120′ (µU/mL)	169.5 ± 141.7	174.2 ± 191.1	ns	152.2 ± 108.1	130.6 ± 55.4	ns	145.2 ± 103.9	229.5 ± 118.7	ns	102.6 ± 70.5	145.5 ± 100.0	ns
	HbA1C (%)	5.54 ± 0.64	5.64 ± 0.78	ns	5.39 ± 0.49	5.7 ± 1.6	ns	5.39 ± 0.46	5.43 ± 0.50	ns	5.48 ± 0.44	5.44 ± 0.70	ns
	HOMA-IR	4.42 ± 3.8	4.49 ± 3.94	ns	5.17 ± 4.54	3.98 ± 2.98	ns	3.73 ± 2.47	3.96 ± 2.96	ns	3.84 ± 3.28	3.01 ± 2.18	ns
	QUICKI	0.323 ± 0.035	0.322 ± 0.033	ns	0.320 ± 0.038	0.326 ± 0.034	ns	0.326 ± 0.030	0.328 ± 0.037	ns	0.340 ± 0.043	0.337 ± 0.032	ns
	ISI Matsuda	3.332 ± 2.93	3.732 ± 3.300	ns	3.207 ± 2.860	4.208 ± 3.215	ns	3.539 ± 2.384	4.498 ± 5.354	ns	4.182 ± 3.171	4.943 ± 1.688	ns
		#/total # (%)	#/total # (%)		#/total # (%)	#/total # (%)		#/total # (%)	#/total # (%)		#/total # (%)	#/total # (%)	
Obesity Classification	Normal (%)	54/233 (23.2)	36/227 (15.8)	0.031	7/36 (19.4) *	7/35 (20.0)	ns	7/20 (35.0)	7/20 (35.0)		10/21 (47.6) *	5/21 (23.8)	
	Overweight (%)	68/233(29.2) ^||^	66/227 (29.1) ^¶^	ns	7/36 (19.4) ^||^	4/35 (11.4) ^¶^	ns	6/20 (30.0)	3/20 (15.0) ^¶^		6/21 (28.6)	11/21 (52.4) ^¶^	
	Obesity (%)	111/233 (47.6)	125/227 (55.1)	ns	22/36 (61.2) *	24/35 (68.6)	ns	7/20 (35.0)	10/20 (50.0)		5/21 (23.8) *	5/21(23.8)	
Metabolic Syndrome Evaluation (ATP III)	Fasting glucose intolerance (%)	34/230 (14.8) ^||^	53/228 (23.2)	ns	5/36 (13.9) ^||^	4/36 (11.1)	ns	4/20 (20.0)	4/20 (20.0)	ns	2/19 (10.5)	4/21 (19.0)	ns
	↑ BP (%)	56/231 (24.2)	58/227 (25.6)	ns	9/36 (25.0)	7/36 (19.4)	ns	3/20 (15.0)	3/20 (15.0)	ns	1/19 (5.3)	4/21 (19.0)	ns
	↑ TG (%)	66/224 (29.5)	71/226 (31.4)	ns	8/35 (22.8)	7/36 (19.4)	ns	6/19 (31.6)	7/20 (35.0)	ns	2/21 (9.5)	3/20 (15.0)	ns
	↓ HDL (%)	135/224 (60.3)	116/226 (51.3)	0.07	25/35 (71.4)	20/36 (55.6)	ns	12/18 (66.7)	9/20 (45.0)	ns	10/21 (47.6)	10/20 (50.0)	ns
	↑ WC (%)	135/230 (58.7)	141/220 (64.1)	ns	27/35 (77.1)	24/34 (70.6)	ns	10/19 (52.6)	9/20 (45.0)	ns	8/21 (38.1)	6/18 (33.3)	ns
	Metabolic syndrome (%)	64/229 (27.9)	59/230 (25.7)	0.015	10/36 (27.8)	10/36 (27.8)	ns	7/20 (35.0)	5/20 (25.0)	ns	2/21 (9.5)	3/21 (14.3)	ns
Evaluation of Glucose Profile	Fasting glucose ≥ 100 mg/dL (%)	31/230 (13.5)	42/228 (18.4)	ns	3/34 (8.8)	3/36 (8.3)		3/20 (15.0)	4/20 (20.0)		1/19 (5.3)	4/21 (19.0)	-
	2 h OGTT intolerance (%)	41/208 (19.7)	14/108(13.0)	ns	2/25 (8.0)	5/20 (25.0)		2/17 (11.8)	1/8 (12.5)		2/13 (15.4)	0/21 (0.0)	-
	HbA1C intolerance (%)	19/173 (11.0)	22/176 (12.5) ^||^	ns	6/29 (20.7)	2/30 (6.7) ^||^		2/12 (16.7)	2/13 (15.4)		2/18 (11.1)	1/17 (5.9)	-
	Total intolerance (%)	91/233 (39.1)	78/233 (33.5)	0.001	11/36 (30.6)	10/36 (27.8)	ns	7/20 (35.0)	7/20 (35.0)	ns	5/21 (23.8)	5/21 (23.8)	ns
T2DM	DM G0′) (%)	0/230 (0.0)	11/228 (4.8)	ns	0/34 (0.0)	1/36 (2.8)		0/20 (0.0)	0/20 (0.0)		0/19 (0.0)	0/21 (0.0)	-
	DM (G120′) (%)	6/208 (2.9)	13/106 (12.3)	ns	3/25 (12.0)	2/20 (10.0)		1/17 (5.9)	2/8 (25.0)		1/13 (7.7)	1/10 (10.0)	-
	DM (HBA1C) (%)	1/173 (0.6)	5/176 (2.8)	ns	0/29 (0.0)	0/30 (0.0)		0/12 (0.0)	0/13 (0.0)		0/18 (0.0)	1/17 (5.9)	-
	Total T2DM (%)	7/233 (3.0)	29/233 (12.4)	0.001	3/36 (8.3)	3/36 (8.3)	ns	1/20 (5.0)	2/20 (10.0)	ns	1/21 (4.8)	2/21 (9.5)	ns
Only OHCC Users			106/233 (45.49)			13/36 (36.11)			10/20 (50)			11/21 (52.38)	ns
OHCC plus MH Drugs Association			34/233 (10.73)			13/36 (36.11)			7/20 (35.6)			5/21 (23.81)	ns

* = *p* < 0.05 phenotypes D and B; †,= phenotype D < A, B, C; ‡ = phenotypes C and B; § = phenotype A > B, C, D; ||= phenotypes A and B; ¶ = phenotype D > A, B, C; # = phenotypes A and D. Mann–Whitney/Wilcoxon; chi-square/Fisher; Tukey paired test; MH = metabolic and hormone. ns= non-significant. ↑ = increased; ↓ = decreased. ns = non significative.

**Table 5 biomedicines-11-03262-t005:** Linear regression between VAI and clinical and biochemical characteristics of the PCOS women according to phenotypes during the follow-up.

		Phenotype A (n = 233) β (95% CI)	*p*-Value	Phenotype B (n = 36) β (95% CI)	*p*-Value	Phenotype C (n = 20) β (95% CI)	*p*-Value	Phenotype D (n = 21) β (95% CI)	*p*-Value
Clinical Characteristics	Follow-up (months)	0.003 (−0.001–0.007)	0.077	0.001 (−0.010–0.011)	0.980	−0.002 (−0.013–0.008)	0.661	−0.006 (−0.017–0.006)	0.285
	F-G Score	0.003 (−0.037–0.044)	0.891	0.005 (−0.099–0.112)	0.909	−0.065 (−0.254–0.125)	0.485	0.234 (−0.085–0.553)	0.129
Hormonal Profile	17OHP *	−0.130 (−0.452–0.192)	0.426	−0.565 (−1.565–0.436)	0.252	−0.015 (−1.563–1.532)	0.983	−0.246 (−3.726–3.234)	0.868
	Testosterone	−0.007 (−0.016–0.001)	0.100	0.003 (−0.028–0.034)	0.854	0.018 (−0.030–0.066)	0.435	−0.087 (−0.145–−0.029)	0.009
	SHBG	−0.005 (−0.010–0.001)	0.103	−0.007 (−0.022–0.009)	0.407	−0.022 (−0.049–0.004)	0.096	−0.035 (−0.058–−0.011)	0.009
	Calculated free testosterone	−0.001(−0.012–0.010)	0.845	0.027 (−0.038–0.091)	0.388	0.064 (0.003–0.125)	0.041	−0.069 (−0.332–0.194)	0.563
	DHEAS	−0.001 (−0.001–−0.001)	0.001	−0.001 (−0.001–0.001)	0.155	0.001 (−0.001–0.001)	0.251	−0.001 (−0.002–0.001)	0.121
	Androstenedione	−0.145 (−0.326–0.039)	0.116	−0.399 (−0.837–0.039)	0.072	0.623 (−0.400–1.646)	0.211	−0.264 (−0863–0.336)	0.333
Obesity Classification	Normal	−1.512 (−2.163–−0.862)	0.001	−1.517 (−3.748–0.714)	0.175	−1.902 (−3.657–0.146)	0.035	−1.478 (−2.529–−0.426)	0.012
	Overweight	0.128 (−0.400–0.656)	0.634	−0.737 (−2.177–0.704)	0.304	0.197 (−0.188–4.182)	0.071	1.178 (−0.528–2.885)	0.150
	Obesity	−0.019 (−0.081–0.042)	0.526	0.082 (−0.119–0.283)	0.398	−0.074 (−0.737–0.590)	0.774	1.152 (−1.285–3.588)	0.307
Metabolic Syndrome Evaluation (ATP III)	Metabolic syndrome	1.671 (1.370–1.972)	0.001	1.369 (0.565–2.173)	0.002	2.772 (1.883–3.661)	0.001	0.323 (−0.828–1.474)	0.536
Diagnosis of Metabolic Changes	Total glucose intolerance	0.925 (0.463–1.388)	0.001	0.736 (−0.469–1.942)	0.221	1.686 (0.234–3.138)	0.025	−0.006 (−1.573–1.561)	0.993
	Total T2DM	2.369 (1.586–3.153)	0.001	1.410 (−0.552–3.372)	0.152	6.409 (3.183–9.636)	0.001	0.885 (−1.625–3.394)	0.440
Metabolic Profile	Total cholesterol	0.007 (−0.001–0.014)	0.053	0.029 (0.012–0.046)	0.002	0.022 (0.005–0.039)	0.013	−0.006 (−0.038–0.025)	0.665
	LDL-c	0.006 (−0.002–0.014)	0.123	0.035 (0.014–0.056)	0.002	0.030 (0.010–0.050)	0.006	−0.002 (−0.038–0.033)	0.882
	G0′	0.032 (0.018–0.046)	0.001	−0.001 (−0.019–0.018)	0.965	0.053 (−0.040–0.145)	0.247	0.014 (−0.017–0.045)	0.325
	G120′	0.019 (0.014–0.024)	0.001	0.019 (0.005–0.033)	0.010	0.026 (0.002–0.051)	0.036	0.007 (−0.052–0.065)	0.772
	I0′	0.037 (0.022–0.053)	0.001	0.020 (−0.030–0.071)	0.421	0.111 (−0.034–2.56)	0.125	0.108 (−0.046–0.626)	0.137
	I120′	0.005 (0.003–0.006)	0.001	0.005 (−0.003–0.013)	0.194	0.017 (0.005–0.029)	0.008	0.024 (−0.051–0.099)	0.154
	HbA1C	0.721 (0.383–1.059)	0.001	0.071 (−0.392–0.535)	0.754	0.320 (−3.006–3.647)	0.836	0.311 (−0.813–1.435)	0.534
	HOMA-IR	0.178 (0.123–0.233)	0.001	0.206 (0.038–0.374)	0.018	0.227 (−0.143–0.597)	0.214	−0.001 (−0.270–0.268)	0.993
	QUICKI	−22.34 (−28.46–−16.21)	0.001	−31.04 (−52.37–−9.71)	0.006	−22.41 (−48.13–3.30)	0.084	−15.27 (−30.06–−0.48)	0.044
	ISI Matsuda	−0.252 (−0.351–−0.153)	0.001	0.0769 (−0.498–0.652)	0.781	−0.331 (−0.619–−0.436)	0.027	−0.256 (−2.096–1.583)	0.327

* Adrenal enzymatic deficiency was excluded by cortrosyn test.

## Data Availability

The data of patient are on the Appendix A.

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
