# Peer review of "Influence of Phenotypes on the Metabolic Syndrome of Women with Polycystic Ovary Syndrome over a Six-Year Follow-Up in Brazil"

_biomedicines, 2023, doi:10.3390/biomedicines11123262_

Round 1

Reviewer 1 Report

Comments and Suggestions for Authors

The paper describes the influence of PCOS phenotypes on metabolic syndrome and on the risk for diabetes mellitus over a six-year treatment period.

The paper is valuable and interesting, I have only some suggestions:

1.      if authors investigate the influence of PCOS phenotypes on the risk for diabetes mellitus generally or type 2 diabetes mellitus, please refine. 

2.      please explain in verse 129 the abbreviation of HOMA-IR

3.      in my opinion the formula of calculation for HOMA-IR, QUICKI and Matsuda should be explain in methods section, not under the tables.

4.      authors should describe with more details the anthropometric parameters such as WC, VIS and explain those abbreviations.

5.      verse 59 correct: (2003)8

6.      all abbreviations should be explain in the text e.g. 17OHP, BP, IGT

7.      improve editorially the whole text, especially the paragraph from verse 131 to 135

8.      correct the fourth part of tables 3 (many ???)

9.      the values in Table 1 are not only present as mean or % but also as median (Q1-Q3)

10.  the colors in figure 2 and 3 are wrong marked - e.g. r= 0.403 is marked using green color instead blue

11. The discussion section would benefit from expansion, and the conclusions should be more closely aligned with both the study's title and its objectives

Author Response

Dear Reviewer,

Thanks for the comments.

Reviewer #1

1) Quality of English Language - I am not qualified to assess the quality of English in this paper if authors investigate the influence of PCOS phenotypes on the risk for diabetes mellitus generally or type 2 diabetes mellitus, please refine.

R: We rewrote in the manuscript “type 2 diabetes mellitus” due to the aim of study was to investigate the influence of PCOS phenotypes on the risk for type 2 diabetes mellitus. We double checked the grammar

2) Explain in verse 129 the abbreviation of HOMA-IR

R: We included the HOMA-IR, Homeostasis Model Assessment-Insulin Resistance. 

3) The formula of calculation for HOMA-IR, QUICKI and Matsuda should be explained in methods section, not under the tables.

R: We included in the methods section and removed from the Tables.

4) Authors should describe with more details the anthropometric parameters such as WC, VIS and explain those abbreviations.

R: We included in the methods sections, the anthropometric parameters: waist circumference (WC) and body mass index (BMI).

5)Verse 59 correct: (2003)8

R: We corrected on the manuscript.

6) All abbreviations should be explained in the text e.g. 17OHP, BP, IGT

R: We included in the methods section: 17-hydroxyprogesterone (17OHP); systemic arterial blood pressure (BP), Impaired glucose tolerance (IGT)

7) Improve editorially the whole text, especially the paragraph from verse 131 to 135

R: We rewrote the paragraph

8) Correct the fourth part of tables 3 (many ???)

R:  We corrected the Table 3

9)  The values in Table 1 are not only present as mean or % but also as median (Q1-Q3)

We corrected Table 1.

10)  The colors in figure 2 and 3 are wrong marked - e.g. r= 0.403 is marked using green color instead blue .

We corrected them.

11)  The discussion section would benefit from expansion.

We improved the discussion.

12) The conclusions should be more closely aligned with both the study's title and its objectives

We rewrote them. 

We highlighted the changes in the manuscript. 

Reviewer 2 Report

Comments and Suggestions for Authors

In the manuscript, entitled »Influence of phenotypes on the metabolic syndrome of women with polycystic ovary syndrome over a six-year follow-up in Brazil «, submitted to Biomedicines for a potential publication, the authors present their research work investigating follow-up of women with PCOS in the context of diabetes mellitus and metabolic syndrome. I am of opinion that in the present form, the article is not good enough to be published, although the topic is interesting. My comments and suggestions to improve it are presented below.

The comments:

1.       In the Introduction the topic has to be presented in more detail with some more studies from the field.

2.       As the topic is well-known, the authors have to clearly state what new knowledge their research brings to the investigated field.

3.       In the first statement the diagnosis of PCOS has to be included.

4.       The four PCOS phenotypes have to be explained as well as the metabolic syndrome components.

5.       Reference for the Ferriman-Gallwey score has to be cited.

6.       The study group has to be presented in more detail in the main text. In addition, the explanation of the recruitment has to be performed more clearly.

7.       The percentage of patients with diabetes as well with metabolic syndrome has to be exposed in the text, at the beginning of the study as well as at the last follow-up.

8.       Was the pharmacological treatment of women unified?

9.       How were non-pharmacological treatment prescribed, standardised and controled?

10.   There is a difference in the age of the study and the control group. Comment on potential bias!

11.   How the insuline tolerance test was performed?

12.   Glucose test and insulin tolerance test were repeted after 5 years. Comment!

13.   Discussion has to be expanded and the results compared with previously published studies.

14.   More than half of the references are older than 10 years!

Comments on the Quality of English Language

mostly satisfactory

Author Response

Dear Reviewer,

Thank you for the comments.

  1. In the Introduction the topic has to be presented in more detail with some more studies from the field.

The well conducted studies on this field is low. We included the details of Carmina and other in this introduction and discussion.

  1. As the topic is well-known, the authors have to clearly state what new knowledge their research brings to the investigated field.

The aim of study to analyze the follow-up of phenotype PCOS over the years in order to the evolution of MS and T2DM. We rewrote it. 

  1. In the first statement the diagnosis of PCOS has to be included.

We include the following paragraph “The NIH consensus panel in 2012 advocated for the sole use of the broader 2003 Rotterdam criteria for diagnosis. It also recommended the identification of the four PCOS phenotypes according to clinical or biochemical hyperandrogenism, ovulatory dysfunction, polycystic ovary morphology”. 

  1. The four PCOS phenotypes have to be explained as well as the metabolic syndrome components.

We included in the methods sections.

  1. Reference for the Ferriman-Gallwey score has to be cited.

The Ferriman-Gallwey score is on the reference [17]: Hatch R, Rosenfield RL, Kim MH, Tredway D. Hirsutism: implications, etiology and management. Am J Obstet Gynecol 1981, 140:815-30.

  1. The study group has to be presented in more detail in the main text. In addition, the explanation of the recruitment has to be performed more clearly.

The recruitment was done by referring patients who came to primary health care in the São Paulo public system.

  1. The percentage of patients with diabetes as well with metabolic syndrome has to be exposed in the text, at the beginning of the study as well as at the last follow-up.

We included in the manuscript: The percentage of T2DM at baseline and at the last follow-up was 3.9% and 11.6%, respectively. The percentage of metabolic syndrome at baseline and at the last follow-up was 29.1% and 25.1%, respectively.

  1. Was the pharmacological treatment of women unified?

The paramacological treatment was based on the clinical and metabolic profile of patient with the standard treatment, such as metformin, statin and anti-hypertensive drugs. We included details on the methods section.

  1. How were non-pharmacological treatment prescribed, standardised and controlled?

All overweight and obese women were advised to start a healthy, hypocaloric diet (the goal daily calorie intake was 1400 kcal) and to engage in daily physical activities, such as walking and aerobic exercises. We controlled the life style parameters in all visits.

  1. There is a difference in the age of the study and the control group. Comment on potential bias!

We included this differences in the limitation of study.

  1. How the insuline tolerance test was performed?

The collect blood for insulin determination with OGTT

  1. Glucose test and insulin tolerance test were repeted after 5 years.

It was mystaped. The OGTT with insulin determination every year.

  1. Discussion has to be expanded and the results compared with previously published studies.

We expanded the discussion.

  1. More than half of the references are older than 10 years!

We done them.

We highlighted the changes in the manuscript.

Reviewer 3 Report

Comments and Suggestions for Authors

See attached 

Comments on the Quality of English Language

minor editing for grammar

Author Response

Dear Reviewer

Thanks for comments

1) The authors present a retrospective analysis of 310 women with PCOS, looking at hormonal, metabolic, and cardiovascular profiles at baseline and then at 6 year follow up. Differences in hormonal and metabolic changes among the 4 NIH PCOS classification groups was analyzed, and it was found that group A was at greatest metabolic and cardiovascular risk. Understanding the progression and cardiometabolic risk in women with PCOS is a very important priority. Strengths of this study include the large sample size and thorough metabolic and hormonal profiling of each patient. However, the main issue is that the analysis does not take into account the treatment(s) received.

R: We included the treatment on the methods: Metformin was prescribed for patients with IR or glucose intolerance. Patients with neither MS nor reproductive desire used a hormonal contraceptive along with antiandrogen drugs for menstrual irregularity and hirsutism.  

2) It is difficult to understand changes in testosterone and glucose, for example, without knowing which women were taking hormonal or metabolic medications.

R: We included the data on the Table 4.

3) The effect of weight on the parameters was also not examined----did the metabolic worsening correlate with weight gain, or was it independent. I’m also not clear on why a control group is presented. The differences noted between PCOS and control at baseline are to be expected and not adding much, and follow up is not presented for the control group to show whether BMI-matched women with PCOS have worse metabolic progression than control women. Would be sufficient to present baseline data on PCOS patients as a whole and then divided into the subgroups and then the follow up as a whole and by subgroup. Consider omitting or decreasing focus on the hormonal changes at follow up, which are almost certainly due to hormonal medication) and focus more on the metabolic/cardiovascular outcomes which are the real point of interest of the manuscript.

R: Regarding weight gain, there was an increasing trend, but it was not significant between the phenotypes (Table 4). We were unable to manage a control group camp (in conditions) to use for a long time, due to lack of patient compliance or adherence for visits. Include this fact in the limitations of study.

4) Specific comments: Abstract-: Add that the patients were divided into groups per NIH recommendations.

R: The patients were divided into four groups per NIH recommendations. We included in the abstract.

5) Methods: Line 59: typo Line 79-82: Were hormone assays they done in morning? Certain time in cycle?

R: The collection was always carried out in the morning. We collected from the 5th to 8th day of the menstrual cycle, in the follicular phase; in amenorrhea, any time, but always taking care to measure concomitant progesterone to exclude collection in the ovulatory phase. We included in methods section.

6) Line 83-84: what type of ultrasound, transvaginal or abdominal, which type of machine was used.

The ultrasound was PowerVision 7000 (Toshiba, Japan) equipped with 3.5 MHz and 7.0 MHz wide-band transducers for abdominal and transvaginal route, respectively. We included in the manuscript.

7) Was this done in office by Gynecology or by radiologist? Same person reading each scan?

The same gynecology read each ultrasound scan. We included in the manuscript.

8) Line 85-86: clarify dose of cosyntropin and cutoff levels used to exclude CAH

We included the dose cosyntropin and cutoff levels on the methods section.

9) Line 88: “the evaluation for tumor”---what was performed?

If the levels of testosterone were greater than 200 ng/dL, CT scan of the upper abdomen or MRI of the pelvis was applied to exclude adrenal or ovarian tumors, respectively. We included in the manuscript.

10) Line 88-89 “ovarian suppression test…” can you explain clarify what results were deemed “insufficient”

The insufficient was when there is no drop of more than 50% of baseline levels in serum total testosterone levels after the test. We included in the manuscript.

11) Line 90-91 Similarly, please clarify what results were consistent with “adrenal component”

The adrenal component was adrenal androgens. We replaced it on the manuscript

12) Line 126: What does VAI stand for?

VAI (visceral adiposity index) is used as a marker for cardiovascular risk related to adipose tissue distribution and function (Amato et al., 2010). We included it in the manuscript.

13) Line 127-128: Please define the phenotypes for the reader who is not familiar with A, B, C, and D phenotypes

We included the definition of the PCOS phenotypes in the methods section: type A – hyperandrogenism, chronic anovulation, and polycystic ovaries; type B – hyperandrogenism and chronic anovulation without polycystic ovaries; type C – hyperandrogenism and polycystic ovaries; and type D – chronic anovulation and polycystic ovaries without hyperandrogenism .

14) Line 134: please state the NCEP-ATP III criteria for readers not familiar

We included in the manuscript in methods section.

15) 138: how was adherence to prescribed drug quantified? By report in the clinic notes?

The adherence was based on the quantification of prescribed drug quantified by the clinic notes in the medical records.  We included it on the methods section.

16) 139: “healthy hypocaloric”---specific Calorie goals and any specific type of diet recommended?

The daily calorie intake was 1400 kcal (healthy hypocaloric). We included in the manuscript.

17) Add to methods: define/clarify the formulas for HOMA-IR, Matsuda ISI and QUICKI Results:

We included them.

18) 164: what is WC? Would add blood pressure measurements

We included them in the methods sections

19) Line 234-241 state the changes in hormonal parameters in women using COCP/anti-androgen but table 3 shows all participants. Please clarify percentage of women taking COCP/antiandrogen in PCOS and In terms of the metabolic parameters, how many taking metformin, statin, etc…? Table 3 contains all patients and would expect differences in those on/off metabolic meds.

We included in the results sections: “Clinical and biochemical evaluations were carried out during the last follow-up visit. The percentage of PCOS patients using medications was the following: a) 45.2% used oral hormonal combined contraceptives (OHCC); b) 19% used OHCC associated with antiandrogens, metformin, or drugs for metabolic correction, such as hypotensors, hypoglycemics, statins, and anorectics; c) 26.1% used metformin alone or in association with statins for metabolic correction; d) 20.6% used no drugs throughout the study due to a reproductive desire or intolerance to metformin or contraceptives; and e) 8.1% used statins, ciprofibrate, levothyroxine, antiandrogens, or psychotropics. The linear regression did not find significant influence on the metabolic profile”.

20) Table 4: please provide for each group the # on hormonal or metabolic medications

We included in the Table 4.

21) Discussion: would talk about the limitations of the 4 group classification (A/B/C/D) and the limited data around whether these classifications are useful for tailoring treatment or predicting outcomes Conclusions: state that “despite metformin treatment” but it is not clarified anywhere which women were taking metformin

We rewrote part of discussion as well as the conclusion. 

We highlighted the changes in the manuscript

Sincerely,

José Maria Soares Júnior, Correspondent author

Round 2

Reviewer 2 Report

Comments and Suggestions for Authors

The revised manuscript, entitled »Influence of phenotypes on the metabolic syndrome of women with polycystic ovary syndrome over a six-year follow-up in Brazil« submitted to Biomedicines for a potential publication, has been substantially improved. The authors have taken most of my  comments  into consideration and I am of opinion, that the revised version can be published.

Comments on the Quality of English Language

mild English editing is requred